

# *TreeSatAI Benchmark Archive*: A multi-sensor, multi-label dataset for tree species classification in remote sensing

Steve Ahlswede[1*], Christian Schulz[2*], Christiano Gava[3], Patrick Helber[4], Benjamin Bischke[4], Michael Förster[2], Florencia Arias[2], Jörn Hees[3], Begüm Demir[1], and Birgit Kleinschmit[2]

[1]Technische Universität Berlin, Germany, Remote Sensing Image Analysis Group, Einsteinufer 17, 10587 Berlin, Germany
[2]Technische Universität Berlin, Geoinformation in Environmental Planning Lab, Straße des 17. Juni 145, 10623 Berlin, Germany
[3]Deutsches Forschungszentrum für Künstliche Intelligenz GmbH (DFKI), Smart Data and Knowledge Services, Trippstadter Str. 122, 67663 Kaiserslautern, Germany
[4]Vision Impulse GmbH, Trippstadter Str. 122, 67663 Kaiserslautern, Germany
[*]These authors contributed equally to this work

**Correspondence:** Christian Schulz (christian.schulz.1@tu-berlin.de), Steve Ahlswede (ahlswedes@gmail.com)

**Abstract.**

Airborne and spaceborne platforms are the primary data sources for large-scale forest mapping, but visual interpretation for individual species determination is labour-intensive. Hence, various studies focusing on forests have investigated the benefits of multiple sensors for automated tree species classification. However, transferable deep learning approaches for large-scale applications are still lacking. This gap motivated us to create a novel dataset for tree species classification in Central Europe based on multi-sensor data from aerial, Sentinel-1 and Sentinel-2 imagery.

In this paper, we introduce the *TreeSatAI Benchmark Archive*, which contains labels of 20 European tree species (i.e., 15 tree genera) derived from forest administration data of the federal state of Lower Saxony, Germany. We propose models and guidelines for the application of the latest machine learning techniques for the task of tree species classification with multi-label data. Finally, we provide various benchmark experiments showcasing the information which can be derived from the different sensors including artificial neural networks and tree-based machine learning methods.

We found that residual neural networks (*ResNet*) perform sufficiently well with weighted precision scores up to 79 % only by using the RGB bands of aerial imagery. This result indicates that the spatial content present within the 0.2 m resolution data is very informative for tree species classification. With the incorporation of Sentinel-1 and Sentinel-2 imagery, performance improved marginally. However, the sole use of Sentinel-2 still allows for weighted precision scores of up to 74 % using either multi-layer perceptron (*MLP*) or Light Gradient Boosting Machine (*LightGBM*) models. Since the dataset is derived from real-world reference data, it contains high class imbalances. We found that this dataset attribute negatively affects the models' performances for many of the underrepresented classes (i.e., scarce tree species). However, the class-wise precision of the best performing late fusion model still reached values ranging from 54 % (*Acer*) to 88 % (*Pinus*). Based on our results, we conclude that deep learning techniques using aerial imagery could considerably support forestry administration in the provision of large-scale tree species maps at a very high resolution to plan for challenges driven by global environmental change.





The original dataset used in this paper is shared via Zenodo (https://doi.org/10.5281/zenodo.6598390) [Schulz et al., 2022]. For citation of the dataset, we refer to this article.

# 1 Introduction

## 1.1 Importance of aerial imagery for forest monitoring

Public and private forest owners in Europe are increasingly confronted with adverse long- and short-term effects on forests driven by global environmental change [IPCC, 2014; MacDicken et al., 2016]. The monitoring of damages caused by events such as storms, heat waves, disease outbreaks and insect infestations has become a highly prominent topics in Earth observation [Holzwarth et al., 2020] which is reflected in the recent remote sensing literature [Tanase et al., 2018; Hollaus and Vreugdenhil, 2019; Senf et al., 2020; Schuldt et al., 2020; Kowalski et al., 2020; Thonfeld et al., 2022]. Thus, long-term mitigation strategies such as the conversion from mono-cultures to more diverse tree stands with higher resilience to environmental pressures have become highly relevant [Hlásny et al., 2017]. To support these strategies, tree species that are most sensitive to recent and upcoming changes must be cost-efficiently mapped and monitored on a large scale.

To date, aerial imagery (i.e., *digital orthophotographs*) taken by plane is the primary large-scale remote sensing data source for forest authorities in Germany. Aerial imagery allows for fieldwork, which is time-consuming and expensive, to be reduced and planned more efficiently. With a spatial resolution in the decimeter range, it also constitutes a considerable source of data for visual check-ups and updates of forest management plans. However, individual species determination through visual interpretation is labour-intensive and requires specialized knowledge by the viewer Hamedianfar et al. [2022].

## 1.2 State of research

In the field of forest remote sensing, aerial imagery is less frequently used than satellite imagery, especially when it comes to forest type classification tasks [Holzwarth et al., 2020]. This may be driven by the low number of spectral bands (red, green, blue, and near-infrared (RGB+NIR)), which is considered to be a limitation in remote sensing applications [Fassnacht et al., 2016; Ganz et al., 2020]. On the other hand, many studies have shown the strengths of multi-spectral satellite imagery for forest classification tasks [Pasquarella et al., 2018; Grabska et al., 2019; Immitzer et al., 2019; Ottosen et al., 2020; Hemmerling et al., 2021; Kollert et al., 2021; Waser et al., 2021; Welle et al., 2022]. The extensive use of the spaceborne data with a much coarser resolution may be boosted by the open data policies of the Landsat and Sentinel programs. However, a higher spatial resolution is an essential factor for improving species prediction [Xu et al., 2021]. For example, on a small spatial extent, studies using RGB imagery from unmanned aerial vehicles with resolutions in the centimeter range have resulted in very high prediction accuracies for species classification using deep learning (DL) techniques [Kattenborn et al., 2020, 2021; Schiefer et al., 2020].



On a large spatial extent, studies using aerial imagery with a resolution in the decimeter range show its potential for *forest cover mapping* [Ganz et al., 2020] and *foliage type classification* [Krzystek et al., 2020] while integrating additional data from LiDAR and multi-spectral sensors. However, few studies have focused on species classification on aerial imagery[Holzwarth et al., 2020].

In summary, the state of research shows the general benefit of *multi-sensor* and *multi-spectral* datasets for forest species classification on all scales. Very high resolution RGB imagery on the centimeter range has also successfully been used for species classification. However, the applicability of similar methods at a larger spatial extent remains unclear. This research gap motivated us to create a benchmark dataset for tree species mapping in Central Europe based on the commonly available aerial imagery and additional freely accessible sensor sources.

## 1.3 Deep learning for large-scale forest applications

Traditional classifiers such as random forest [Breiman, 2001] or support vector machines [Boser et al., 1992] have been commonly used for classifying tree species using multi-spectral images [Sesnie et al., 2010; Karlson et al., 2015]. One drawback of such approaches is their reliance on handcrafted features (i.e., *expert rules*) which require domain knowledge and often fail to include all pertinent features [Wurm et al., 2019]. DL-based models are able to overcome this requirement as they automatically learn large sets of features which are specific to the given classification targets. Thus, DL models have attracted great attention for tree species classification [Egli and Höpke, 2020; Schiefer et al., 2020; Martins et al., 2021; Zhang et al., 2021]. However, DL models require large amounts of labeled data in order to sufficiently learn the optimal model parameters, leading to bottlenecks in its application Hamedianfar et al. [2022]. One possible solution to this problem is to use networks which have been pre-trained on large datasets such as ImageNet [Deng et al., 2009], and then training the network on the smaller target training set afterwards. While this approach does indeed improve performance over training a model from scratch on small training datasets, recent work has shown that pre-training the network on imagery which is more closely related to the target dataset leads to a better performance [Sumbul et al., 2019].

DL datasets for large-scale land use and land cover applications often provide forests as a single class (e.g., *UC Merced* [Yang and Newsam, 2010], *Deepsat* [Basu et al., 2015]), and *EuroSAT* [Helber et al., 2019] or as multiple classes based on foliage, seasonality and density types (e.g., *BigEarthNet* [Sumbul et al., 2019], *SEN12MS* [Schmitt et al., 2019]). The *NeonTreeCrowns* [Weinstein et al., 2020] and *NEON Tree Evaluation* datasets [Weinstein et al., 2019a] represent the first large-scale DL datasets exclusively related to trees. They include data from multiple sensors including LiDAR, RGB, aerial, and hyperspectral imagery across different forest types in the United States. With *DeepForest*, the authors also proposed a semi-supervised DL neural network for individual tree crown delineation using airborne RGB data [Weinstein et al., 2019a, b]. However, DL datasets for *tree species-level* classification on a large-scale have not yet been published. Therefore, we believe that the research community could greatly benefit from a dataset upon which to pre-train DL models, given that the model will have already learned features relevant for tree species.



### 1.4 Study aim

Given the potential improvement in performance from high-resolution aerial imagery and additional spectral information from Sentinel sensors, we introduce the *TreeSatAI Benchmark Archive* [Schulz et al., 2022]. Based on reference data from forest administration data in Germany, the dataset aims to gather multi-sensor and multi-label information for the classification of 20 tree species at Central Europe. The *TreeSatAI Benchmark Archive* consists of 50,381 image patches from aerial, Sentinel-2 (S2), and Sentinel-1 (S1) imagery (Fig. 1), with a range of 212 to 6,591 individual samples per class.

In this paper, we propose guidelines for the application of the latest machine learning techniques in forest remote sensing while handling an unweighted (i.e., *real-world*) dataset. In order to test the applicability of the multi-sensor imagery for species classification, we provide various benchmark experiments showcasing the information which can be extracted via various DL models. Specifically, we investigate the use of each sensor source individually, as well as multi-sensor fusion models examine the applicability of all imagery types present within the *TreeSatAI Benchmark Archive*.

The decision on the integration of multi-sensor data follows the state of research which has shown that a large range of spectral information can be important to classify species at finer level based on its wavelengths. This could provide important discriminating information for classifying species with similar spatial and spectral features within the high-resolution RGB+NIR aerial data.

## 2 Dataset description

### 2.1 Study area

The *TreeSatAI Benchmark Archive* contains image data covering publicly managed forests in the federal state of Lower Saxony, Germany. The study area comprises both flat lands with a maritime climate (i.e., wet and winter-mild) in the North-West as well as low mountain ranges with a more continental climate (i.e., dry and winter-cold) in the South-East [Beck et al., 2018]. Almost 22 % of the federal state consists of forests, of which large proportions belong to the continental parts of the study area [Holzwarth et al., 2020]. According to NW-FVA [2021], the five predominant tree species at our research area were pine (*Pinus sylvestris*) with 38 %, beech (*Fagus sylvatica*) with 16 %, spruce (*Picea abies*) with 13 %, and oak (*Quercus robur*, *Q. petraea*) with 7 % of the forests. These species are usually grown as pure stands. The remaining tree stands cover 26 % of the research area. Among many others, they contain species such as ash (*Fraxinus excelsior*), Douglas fir (*Pseudotsuga menziesii*), larches (*Larix decidua*, *L. kaempferi*) and cherry (*Prunus avium*).

### 2.2 Reference data collection

For label extraction, we used forest administration data provided by the Lower Saxony State Forest Management Organisation [NLF, 2021a, b] (Fig. 2). The rolling archive is annually updated through spatially rotating terrestrial surveys and flight campaigns [Böckmann, 2016]. It contains ground truth data and aerial imagery from 2011 to 2020, both corresponding to the same years.

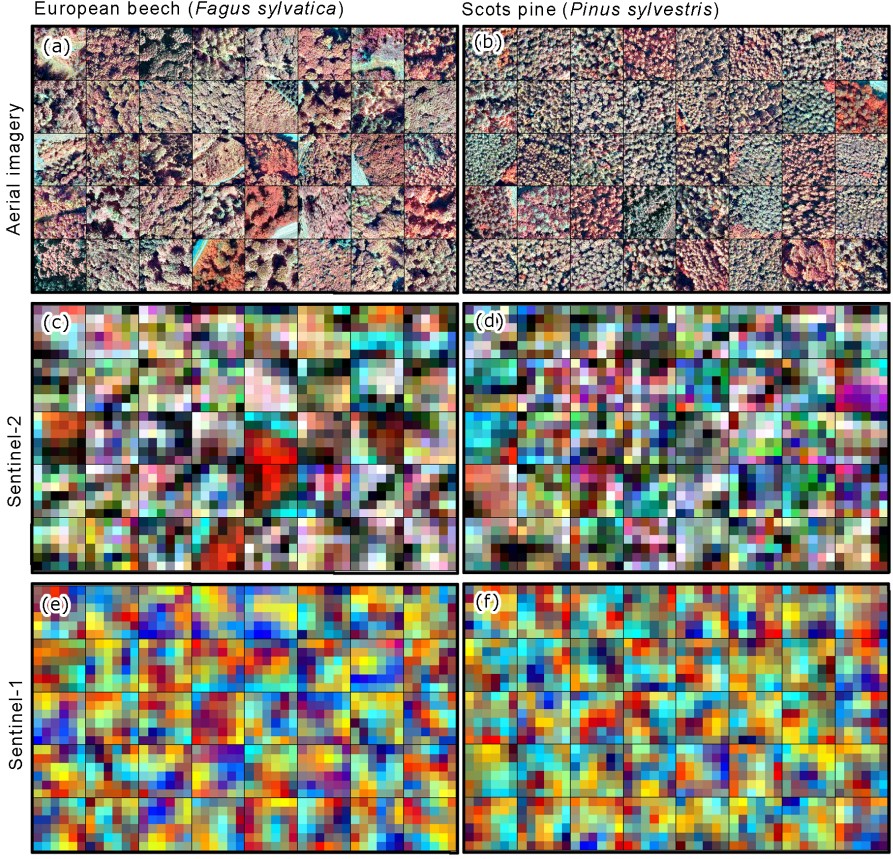

**Figure 1.** Examples of 60x60 m image patches from the *TreeSatAI Benchmark Archive* for two of the 20 species. (a+b) Aerial imagery from 2011–2020 with 0.2 m resolution (near-infrared, green, and blue). (c+d) Sentinel-2 imagery 2015–2020 with 10 m resolution (near-infrared, green, and blue). (e+f) Sentinel-1 imagery 2015–2020 with 10 m resolution (VV, VH, and VV-VH cross ratio). The images contain modified data from the NLF 2021a, b and the ESA 2021.

Two forest administration datasets were chosen for label extraction. First of all, from the *federal state forest management data* (Waldeinrichtungsflächen der Niedersächsischen Landesforsten, *WEFL*), a subset of 175,142 vector geometries was made available. The dataset is derived from administrative surveys and interpretation of aerial imagery and collects information on the stand type and age class for single forest stands. Second, from the *federal state forest inventory* (Betriebsinventur der Niedersächsischen Landesforsten, *BI*), a subset of 18,289 point geometries was made available. The dataset covers permanent ground truth points with a 100 m distance grid over the study area [Saborowski et al., 2010]. For a 13 m radius around each point, the *BI* collects multiple information on the tree stand including the main tree species, stand type, and stand age.

With the *stand type*, both datasets share a common attribute for derivation of tree species labels. However, both datasets have strengths and weaknesses in labeling quality and potential sample numbers. While having a very precise annotation from ground truth, the number of point features is limited in the *BI* data. In comparison, the high number of geometry features in

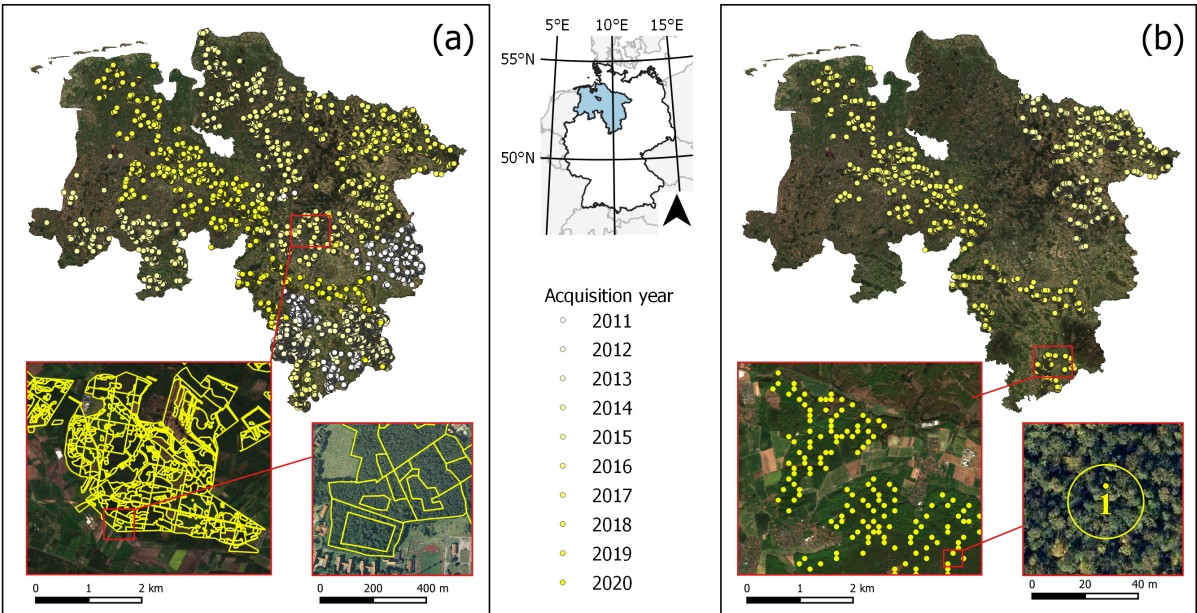

**Figure 2.** Spatial coverage of the *federal state forest management data* [NLF, 2021b] (a) and *federal state forest inventory* [NLF, 2021a] (b) of Lower Saxony used for label derivation. The background imagery contains modified Sentinel-2 data from the ESA provided by the LGLN [2020].

the *WEFL* data allows for generating large training datasets. Additionally, larger geometries can be used for the extraction of multiple samples for underrepresented classes. However, missing ground checks can in some cases lead to outdated or mislabeled information.

## 2.3 Remote sensing data collections

### 2.3.1 Aerial image collection

Aerial imagery from multiple flight campaigns from midsummer was provided with the forest administration data. The image collection covers four spectral bands (RGB+NIR) and has a spatial resolution of 0.2 m. With all the flight campaigns from the years 2011 to 2020, a full coverage of the study area is reached. The aerial imagery and the forest administration data have the same acquisition year and thus correspond to each other.

### 2.3.2 Sentinel-1 image collection

Especially by linking the backscatter information to tree crown volume and density [Cherrington et al., 2019], Synthetic Aperture Radar (SAR) data could potentially be useful for tree species classification. Thus, we collected S1 scenes from mid-August over Lower Saxony for the years 2015 to 2020. The C-Band SAR sensor has a resolution of 10 m and a revisit time



of less than 6 days [Torres et al., 2012]. The S1 scenes were pre-processed into 3-channel georeferenced image mosaics. The resulting images consist of VV and VH polarized channels and their ratio in the third channel.

As input for the processing pipeline, we used Level-1 Ground Range Detection (GRD) products in interferometric Wide
Swath (IW) mode which were provided by the European Space Agency (ESA). With the *S1TBX - ESA Sentinel-1 Toolbox* from the Sentinel Application Platform (SNAP), we performed the following processing steps: We started with the application of orbit files, followed by border noise removal and radiometric calibration. For the orthorectification of the images, we applied range doppler terrain correction with the DEM SRTM 30 for scenes below 60 degrees and ASTER DEM otherwise. We converted the backscatter values from linear units to the decibel scale. As the last step, we updated the resolution of the
imagery to 10 m using bicubic interpolation to match the spatial resolution of the S2 image collection.

### 2.3.3   Sentinel-2 image collection

To correspond with the other image collections temporally, we collected the S2 scenes over Lower Saxony for the years 2015 to 2020. The multi-spectral imagery of S2 contains surface and atmospheric reflectance values including 13 bands in the visible and infrared range with a spatial resolution of 10 to 60 m [Drusch et al., 2012].
We atmospherically corrected all S2 Level L1C scenes using *SNAP supported plugin Sen2Cor_v2.9* and resampled all bands to 10 m resolution. Both the data and software are provided by the ESA. The cirrus band was omitted due to lack of information for most land surfaces. Although S2 imagery is collected by two satellites with a repetition rate between three and five days, multiple scenes from the same month can be affected by clouds. Additionally, for the years 2015 and 2016, only one of the two S2 satellites were in orbit. In order to cope with having observations from only one satellite and high cloud coverage, we
computed cloud-free mosaics for the summer of each year with median filtering over scenes from the months July, August, and September.

### 2.4   Sampling strategy

The goal of the sampling strategy was to generate highly reliable labeled image patches from all three image collections. This means that the timestamp of all patches must fit to the respective label year. For labels older than 2014–when there was no S1
and S2 data available–we generated samples from the 2015 data to ensure annotations with the shortest temporal distance. To ensure the maximum number of image patches, we took both datasets *WEFL* and *BI* for label extraction.

Then, multiple sampling steps were done to derive single-labeled image patches (Fig. 3): (1) *Class selection and aggregation*: Based on the attribute *stand type* with 64 classes, we selected all classes with *pure tree stands*. If a pure class was not available for a tree species, we chose the respective *pure and mixed* or *mixed* classes and aggregated them. All other classes were
excluded. After this step, 20 classes remained, which were re-labeled by the scientific name of the main tree species (Fig. 4). (2) *Polygon sample selection* (only *WEFL*): We excluded all polygons smaller than 2,000 m² to avoid the generation of image patches from very small tree stands. (3) *Sample point creation* (only *WEFL*): Within the polygons of each class we created 5,000 random points. They are supposed to be the centre of the image patches. If possible, a negative buffer of 20 m to the polygon boundary was set to reduce the proximity of the points to neighboring tree stands. (4) *Exclude points by distance*



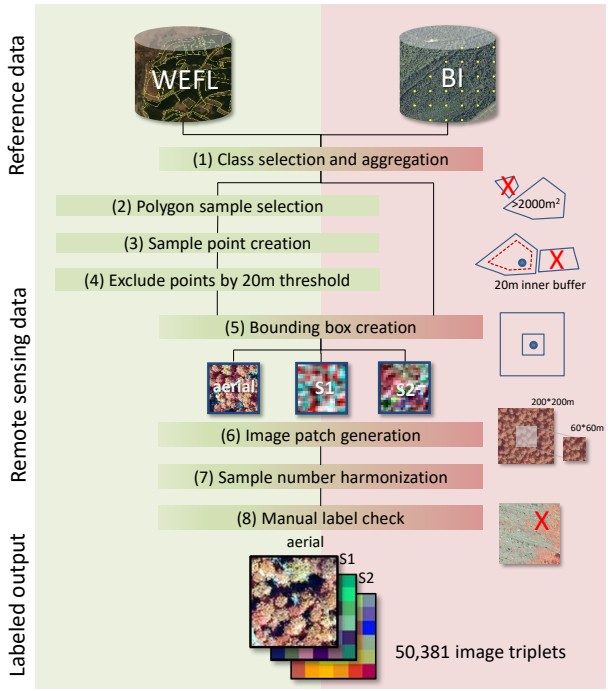

**Figure 3.** Flowchart of the sampling strategy for the patch generation (aerial, Sentinel-1 and Sentinel-2) from federal state forest management data (WEFL) [NLF, 2021b] (A) and federal state forest inventory (BI) [NLF, 2021a] (B) of Lower Saxony.

*threshold* (only *WEFL*): To reduce the overlap of image patches, we excluded sample points with less than 20 m distance from each other. (5) *Bounding box creation*: For each of the sample points a bounding box was created which depicts the size of the image patches. We chose 60x60 m bounding boxes leading to *300x300 pixels* patches for the aerial imagery and *6x6 pixels* patches for the S1 and S2 imagery. (6) *Image patch generation*: From the three image collections, we extracted the image patches according to the bounding boxes. Each file was named according to its genus, the species, the age class, the

unique sample ID, the dataset, and the dataset source (e.g., *Fagus_sylvatica_6_154201_BI_NLF.tif* ). For alternative labeling, we refer to the additional classes in Figure 4. For multi-labeling files, we refer to the description of the Zenodo archive. (7) *Sample number harmonization*: Because of data gaps in the image collections, mainly caused by clouds in the S2 imagery, a large proportion of incomplete image triplets had to be removed. (8) *Manual label check*: From the remaining image patches, a minor number was wrongly labeled (e.g., different species) or had other issues (i.e., missing bands, haziness, blurriness etc).

These images were manually removed through visual check-ups.




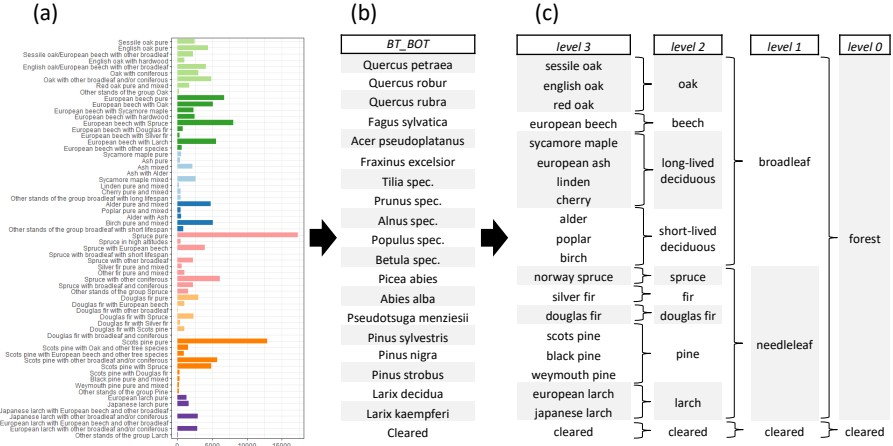

**Figure 4.** Original stand type classes from the forest management data (A) which were used to derive *TreeSatAI Benchmark Archive* single-labels on the species-level (B). The optional labels (C) can be used for re-labeling and include the English name (level 3), forest management classes (level 2), leaf types (level 1) and land cover classes (level 0).

## 3 Benchmark models

In order to provide some initial benchmarking of the presented dataset and to answer fundamental questions in the context of forest remote sensing, we evaluate and compare recent machine learning methods for uni- and multi-sensor scene classification. We primarily focus on DL models given the main aim of this large-scale dataset is to enable the application of those methods for tree species classification. However, we also investigate a Light Gradient Boosting Machine (LightGBM) [Ke et al., 2017] in order to provide a comparison of tree-based machine learning methods with DL.

### 3.1 Deep learning models

For the aerial data, we apply a ResNet18 [He et al., 2016] in order to model the spatial and spectral content. This model has shown high performance across various image processing tasks such as image classification [Scott et al., 2017], semantic segmentation [Chen et al., 2018], and content-based image retrieval [Sumbul et al., 2021]. We assess training the network from scratch versus fine-tuning a pre-trained network. For the pre-trained network, we use model parameters obtained from training on ImageNet. Given that the ImageNet pre-trained models expect three bands as input, we also evaluate different configurations: 1) using only three of the possible four aerial bands, and 2) duplicating one of the input channels in the pre-trained network in order to allow the use of all four aerial bands as input.

To model S1 and S2 data, we examine three different model architectures (Fig. 5). Given that the image patches are only 6x6 pixels in dimension, a standard 2-D convolutional network makes little sense as the spatial area to convolve is small. As such, we investigate three models which focus more on spectral rather than spatial content. The first model is a simple fully connected network in which each pixel in the S1 and S2 data is fully connected to each hidden node via a linear layer (Fig. 5A).

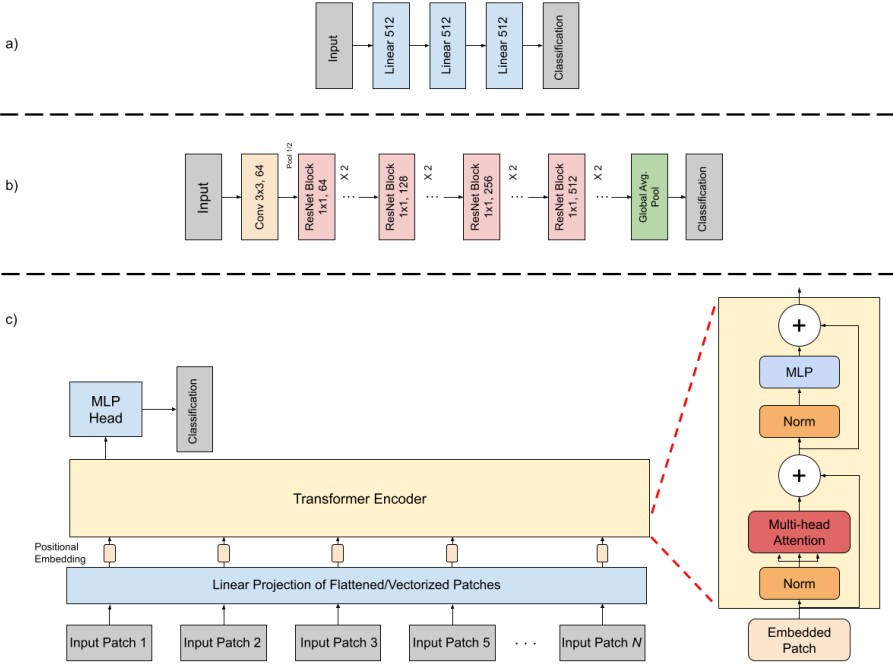

**Figure 5.** The architecture of a) multi-layer perceptron (MLP), b) 1-D residual neural network (ResNet), and c) Vision Transformer which were applied for Sentinel-1 and Sentinel-2 data.

The second model is a residual neural network (ResNet) utilizing 1-D convolutions instead of the standard 2-D. Convolutional
neural networks (CNNs) using 1-D convolutions are often applied in hyperspectral classification problems [Ansari et al., 2021]
where modelling spectral content, rather than spatial content, is the aim. We then test a 1-D ResNet [Hannun et al., 2019] for
modelling the S1/S2 data (Fig. 5B). The final model is a vision transformer (ViT) [Dosovitskiy et al., 2020]. The ViT performs
image classification by breaking an image down into small patches which are used as input tokens. The information within
each token, as well as the relationships between the tokens, is learned by the model. Given the small sizes, our Sentinel images
need not be split into patches, and thus we use each individual band of the image as an input token. In this way, the model
should learn the information within each band, while simultaneously learning the relationships between the bands as well (Fig.
5C).

To jointly model S1 and S2 data with the aerial data, some form of fusion technique is required in order to combine the multi-
sensor data. A naive approach is to simply concatenate the data along the channel dimension and then training a classifier (e.g.,
CNN). Such an approach is known as early fusion, given that the data is fused at the earliest possible stage. However, given
the large discrepancies in spatial resolution between the aerial and S1/S2 data, concatenation at the image level is not possible
without applying large amounts of interpolation to the Sentinel data. Late fusion is another technique which has shown greater
performance than early fusion [Hong et al., 2020]. Here, each modality is passed through its own separate network, and the
fusion is performed on the final features. Such an approach allows for differences in the sizes of images from the various

modalities since fusion is done once a single feature vector for each modality has been extracted. Thus, we focus our multi-sensor learning experiments on late fusion. In these experiments, the aerial data is modelled using a ResNet, whereas the S1 and S2 branches each use one of the models mentioned above (Fig. 5).

### 3.2 Tree-based machine learning models

Another aspect we explore is the benefit of DL methods over traditional machine learning approaches. To this end, we use
LightGBM [Ke et al., 2017], a framework for machine learning comprising several decision tree algorithms, including classification. Training DL models is energy and time consuming. In contrast, LightGBM provides algorithms for classification that are fast, memory efficient, and can be executed in parallel. In other words, the cost of using a LightGBM classifier is significantly lower than that of training a deep model. Consequently, it is relevant to measure the effective accuracy boost of deep models in comparison to standard machine learning classifiers.

LightGBM classifiers were designed to solve single-label problems. Thus, since the *TreeSatAI Benchmark Archive* is a multi-label dataset, we cannot directly use LightGBM algorithms for tree species classification. To circumvent this, we employ a heuristic method known as One-versus-Rest (OvR). In our scenario, OvR is used to cast a multi-label classification problem into multiple binary classification problems, one for each class.

For the classification of aerial images with LightGBM, we first use a ResNet18 pre-trained on ImageNet to extract features
and then use these features to train the decision tree classifier. Here, we ignore the NIR band. Regarding fusion of aerial with S1 and/or S2, we concatenate the features predicted by the ResNet18 on the aerial imagery with data from S1, S2 or both and then train the LightGBM classifier.

## 4 Experimental results

In our experiments we aim to answer five main questions with respect to the dataset. The first question examines which of
the three sensors is most suitable for the classification, as we have aerial data with high spatial resolution and Sentinel data with a high range of spectral information. The second question is how to best train the model for aerial data. Here, we test three strategies for training our ResNet model: 1) using a model pre-trained on ImageNet data and retraining only the final classifier layer; 2) fine-tuning the weights of the final layer and classifier of a ResNet pre-trained on ImageNet; and 3) training the model from scratch. The third question is whether the inclusion of the NIR band brings a significant benefit to the model
performance. This is examined for two cases: 1) training from scratch; and 2) fine-tuning a pre-trained model. Given that the model pre-trained on ImageNet typically only allows three bands as input, we duplicate the third input channel's weights when including the NIR band. The fourth question is to examine whether the fusion of aerial with Sentinel data improves the model performance compared to using only aerial data. As mentioned above, we use three different models in a late fusion setting for this. The final question we investigate is the ability of the various models to classify under-represented classes within the
dataset.



For all experiments, we trained models for a total of 150 epochs using a batch size of 32. For the learning rate, we use a cyclic learning rate scheduler [Smith, 2017] which modulates the learning rate value between 0.00005 and 0.001.

## 4.1 Results of using individual input sources

Here, we examine the three data modalities of aerial, S1 and S2 available in the *TreeSatAI Benchmark Archive*. Table 1 shows that from the ResNet and Multi-layer perceptron (MLP) models the best performance is achieved when using the aerial data, followed by S2 and then S1. In terms of weighted $F_1$ and mAP, the models trained using aerial data outperformed those trained with S2 by 17 % and 13 %, respectively. For the micro $F_1$ and mAP, using aerial data outperforms S2 by 15 % and 12 %, respectively. This indicates that the spatial content present within the aerial data is very informative for the classification of the various tree species. The lower S2 and S1 scores are likely because of the lower spatial resolution of 10m compared to aerial with 0.2m per pixel. However, the larger number of spectral bands in S2 still allows for mAP scores of about 65 % and $F_1$ of around 53 %. When examining the scores obtained using S1 data, we can see that using the 10m resolution back-scattering information alone may not be sufficient for tree species classification.

The LightGBM model results show similar scores for S1 and S2. However, for the aerial data the model reaches lower weighted $F_1$ and mAP scores with losses of 25 % and 17 % respectively. This indicates that extracting features with a ResNet18 and using these features to train a decision tree classifier is not recommended for aerial images.

**Table 1.** Classification performances using single modalities of either aerial, Sentinel-2, or Sentinel-1 data.

| Model | Data source | Weighted | | | | Micro | | | |
|---|---|---|---|---|---|---|---|---|---|
| | | Precision | Recall | $F_1$ | mAP | Precision | Recall | $F_1$ | mAP |
| ResNet | Aerial | **79.18** | **62.18** | **69.47** | **77.43** | **79.54** | **62.18** | **69.80** | **78.25** |
| MLP | Sentinel-2 | 74.59 | **42.23** | **51.97** | **64.19** | 77.18 | **42.23** | **54.59** | **65.83** |
| MLP | Sentinel-1 | 33.29 | 7.13 | 10.09 | 29.42 | **63.01** | 7.13 | 12.82 | 33.09 |
| LightGBM | Aerial | 76.12 | 33.98 | 43.99 | 60.04 | 78.63 | 33.98 | 47.46 | 57.98 |
| | Sentinel-2 | 74.27 | 40.04 | 48.17 | 61.99 | 76.27 | 40.04 | 52.52 | 61.66 |
| | Sentinel-1 | 37.96 | 8.06 | 11.86 | 32.79 | 55.49 | 8.06 | 14.07 | 35.11 |

## 4.2 Effect of near-infrared band

In this section, we examine the influence of including the NIR band when training the network on the aerial data. We tested this across three training scenarios: 1) training the model from scratch (*Scratch*); 2) fine-tuning all layers of a pre-trained network (*Fine-tuning All*); and 3) fixing all the convolutional layer weights from a pre-trained network and only training the classification head of the model (*Retrain Head*). Pre-trained models used weights obtained by training on ImageNet.

As seen in Table 2, the NIR band does help when we train from scratch, as both the micro and weighted $F_1$ and mAP scores all increased by 2 % compared to using only RGB inputs. This is to be expected as the NIR band is an important indicator for vegetation monitoring [Tucker, 1979].

When we look at both variations of the pre-trained models, the result is the opposite as the $F_1$ and mAP scores are higher when using only the RGB channels. This is likely due to both pre-trained models having been pre-trained on RGB inputs. As





such, the duplicated channel used for NIR in the pre-trained scenarios came from a channel which was not originally optimized for NIR input. Given that the value ranges of the NIR channel are higher than those of any of the RGB channels, the additional duplicated channel negatively influenced the pre-trained model performances. The three band *Fine-tuning All* variant obtained 1 % metric improvements over the four band variant in $F_1$ and mAP. Likewise, the three band *Retrain Head* outperformed 275 the four band variant by 6–11 % in the $F_1$ and mAP metrics. For the four band *Retrain Head*, the duplicated channel was not retrained for the NIR input and thus the model was unable to adapt the network weights for the NIR input, leading to larger discrepancies between the three and four band performances. The *Fine-tune All* model was better able to adapt the duplicated channel to the NIR input as its pre-trained weights were further trained.

**Table 2.** Model classification performances using aerial data when training from scratch (*Scratch*), using a frozen pre-trained (ImageNet) ResNet backbone and training a classification head ontop (*Retrain Head*), and fine-tuning all the layers of a pre-trained (ImageNet) ResNet (*Fine-tuning All*). Performances for each scenario are given without (RGB) and with the near infrared band (RGB+NIR).

| Scenario | NIR | Weighted | | | | Micro | | | |
|---|---|---|---|---|---|---|---|---|---|
| | | Precision | Recall | $F_1$ | mAP | Precision | Recall | $F_1$ | mAP |
| Scratch | X | 78.88 | 59.27 | 67.37 | 75.87 | 79.05 | 59.27 | 67.75 | 76.49 |
| Scratch | ✓ | **79.18** | 62.18 | 69.47 | 77.43 | **79.54** | 62.18 | 69.80 | **78.25** |
| Fine-tuning All | X | 78.09 | 64.35 | **70.05** | **78.37** | 76.31 | 64.35 | **69.82** | 76.98 |
| Fine-tuning All | ✓ | 75.82 | **64.84** | 69.43 | 77.31 | 74.75 | **64.84** | 69.44 | 75.72 |
| Retrain Head | X | 70.11 | 22.57 | 31.66 | 49.13 | 73.17 | 22.56 | 34.50 | 48.46 |
| Retrain Head | ✓ | 46.47 | 19.55 | 22.38 | 39.20 | 53.16 | 19.56 | 28.60 | 37.25 |

### 4.3 Analysis of different training strategies for aerial data

In this section, we discuss whether using a pre-trained network performs better than training the network from scratch. We thus test the same three training scenarios as in the previous sub-section, but here we focus only on comparing the best model from each scenario. In Table 2, we see that the *Fine-tuning All* network leads to slightly improved metric performance when compared to *Scratch*, while the *Retrain Head* scenario performed the worst. Comparing the *Fine-tuning All* with *Scratch*, we see roughly 1 % performance increases for both weighted $F_1$ and weighted mAP, while the micro average of $F_1$ was equal. 285 Comparing these two models further, we see the micro mAP was higher by 1 % for *Scratch*, as well as the weighted and micro precisions, which were higher than the *Fine-tuned All* by 1 % and 3 %, respectively. This suggests that training from scratch can achieve nearly identical performance as using a pre-trained network. The *Retrain Head* variant showed a very low performance, achieving roughly half the performance as the *Fine-tuning All* variant for both $F_1$ and mAP scores. This indicates that the features learned from training on ImageNet need fine-tuning when being applied to remote sensing image classification 290 tasks. Given that the pre-trained network does slightly outperform the network trained from scratch, we apply the three band fine-tuned network for the aerial data in all multi-sensor learning approaches.

## 4.4 Results of multi-sensor fusion

Although the earlier results from Table 1 indicate that the information from S1 and S2 is weaker than that of the aerial data,
the combination of the various modalities may provide a greater benefit than their individual use. Thus, we examine the
performance of various multi-sensor learning techniques. Table 3 shows the results for different models applied to the multi-
sensor fusion of aerial and Sentinel data. We see that in terms of $F_1$ and mAP scores, simple MLP performs best, outperforming
the 1-D CNN and Transformer variants by about 1 % across both micro and weighted $F_1$ and mAP. This slight increase in metric
performance may be due to the full connections within the MLP network being better suited to modelling all of the information
present within the small Sentinel patches.

All DL-based methods outperformed the LightGBM baseline approach in terms of $F_1$ and mAP scores. However, the Light-
GBM did manage to obtain the highest weighted precision score using the combination of aerial and S2, and the highest micro
precision using the combination of aerial, S1 and S2. On the other hand, the mAP scores for the LightGBM was about 10 %
lower than the other fusion models, indicating that LightGBM performed well at the 50 % threshold for a positive prediction,
but had lower performance across the various thresholds calculated in the mAP.

**Table 3.** Classification performances of various models used for multi-sensor fusion of aerial (A), Sentinel-2 (S2), and Sentinel-1 (S1) data.

| Models | | Weighted | | | | Micro | | | |
|---|---|---|---|---|---|---|---|---|---|
| | | Precision | Recall | $F_1$ | mAP | Precision | Recall | $F_1$ | mAP |
| ResNetPT + LightGBM | A + S1 | 75.52 | 34.47 | 44.52 | 58.22 | 76.97 | 34.47 | 47.62 | 57.11 |
| | A + S2 | **80.89** | 44.86 | 54.40 | 69.08 | 81.46 | 44.86 | 57.86 | 69.41 |
| | A + S1 + S2 | 80.09 | 44.42 | 54.26 | 68.85 | **81.72** | 44.42 | 57.55 | 69.62 |
| ResNetPT + MLP | A + S1 | 79.52 | 65.24 | 71.45 | 79.09 | 78.96 | 65.24 | 71.45 | 78.71 |
| | A + S2 | 80.27 | 63.50 | 70.55 | 79.25 | 79.63 | 63.50 | 70.66 | **79.20** |
| | A + S1 + S2 | 79.61 | 65.36 | **71.54** | **79.50** | 79.29 | 65.36 | **71.66** | **79.20** |
| ResNetPT + 1-D CNN | A + S1 | 76.15 | 65.60 | 70.29 | 77.19 | 75.49 | 65.60 | 70.20 | 76.35 |
| | A + S2 | 77.64 | 66.12 | 71.17 | 78.24 | 76.88 | 66.12 | 71.10 | 79.00 |
| | A + S1 + S2 | 76.53 | **66.21** | 70.88 | 78.28 | 76.03 | **66.21** | 70.78 | 77.40 |
| ResNetPT + Transformer | A + S1 | 80.32 | 62.97 | 70.06 | 78.58 | 79.02 | 62.97 | 70.09 | 78.07 |
| | A + S2 | 79.85 | 64.03 | 70.74 | 79.07 | 79.24 | 64.03 | 70.83 | 78.85 |
| | A + S1 + S2 | 79.50 | 63.62 | 70.20 | 79.07 | 78.59 | 63.62 | 70.32 | 78.67 |

When compared to the uni-sensor approach which only utilizes aerial data, the incorporation of S1 and S2 data provides an
increase of 1 % for the weighted $F_1$ and mAP, and a 2 % increase in the micro $F_1$ and mAP. Thus, the high resolution aerial
imagery provides the most important features for differentiating the various species, while the spatial resolution of S1 and S2
data is likely too low to provide a larger benefit. Other works [Immitzer et al., 2012] have shown contradictory results, where
the addition of multi-spectral bands outside of RGB+NIR led to large benefits in the classification of trees at the species level.
However, that study was done using WorldView-2, and in that case the spatial resolution of the additional multi-spectral bands
was higher (0.5 m) than that of the Sentinel data used in our study.



## 4.5 Learning under-represented classes

Here, we examine the class based performances using the best performing model (multi-sensor late fusion using MLP, Table 4). The model performance is highest for classes *Pinus* (82 % $F_1$, 73 % mAP), *Picea* (79 % $F_1$, 69 % mAP), *Abies* (77 % $F_1$, 61 % mAP) and *Quercus* (71 % $F_1$, 60 % mAP). The model performed worst when predicting *Tilia* (47 % $F_1$, 23 % mAP), *Prunus* (52 % $F_1$, 31 % mAP), *Acer* (53 % $F_1$, 31 % mAP), *Betula* (54 % $F_1$, 35 % mAP) and *Fraxinus* (58 % $F_1$, 36 % mAP). The majority of the classes for which the model was most successful corresponded to the most frequently-occurring classes, whereas the classes for which the model performed poorly were those with few image samples. This class imbalance mirrors the real-world situation at our study site. The one exception was the class *Abies*, which made up only 2 % of the image samples, but had the third highest classification scores. This result suggests that the tree stands with silver fir (*Abies alba*) contain some easily identifiable features that set it apart from other classes.

**Table 4.** Class-wise performances using the optimally performing late fusion multi-sensor model. For this model, aerial data was passed through a ResNet, and Sentinel-1 and Sentinel-2 were passed through a multi-layer perceptron.

| **Species** | Precision | Recall | $F_1$ | mAP | Support |
|---|---|---|---|---|---|
| Abies | 82.73 | 73.39 | 77.78 | 61.37 | 1002 |
| Acer | 53.99 | 53.66 | 53.82 | 31.98 | 2517 |
| Alnus | 66.30 | 55.62 | 60.50 | 39.78 | 2598 |
| Betula | 71.43 | 43.41 | 54.01 | 35.02 | 2675 |
| Cleared | 78.54 | 68.43 | 73.14 | 57.29 | 4362 |
| Fagus | 84.10 | 60.68 | 70.50 | 59.94 | 8482 |
| Fraxinus | 60.45 | 56.84 | 58.59 | 36.80 | 2301 |
| Larix | 71.37 | 69.04 | 70.19 | 52.28 | 3706 |
| Picea | 85.91 | 74.75 | 79.94 | 69.73 | 8475 |
| Pinus | **87.79** | **77.50** | **82.33** | **73.22** | 8822 |
| Populus | 83.05 | 55.05 | 66.22 | 46.52 | 391 |
| Prunus | 78.95 | 39.47 | 52.63 | 31.62 | 301 |
| Pseudotsuga | 79.61 | 63.33 | 70.54 | 53.69 | 3406 |
| Quercus | 83.21 | 62.90 | 71.64 | 60.81 | **9344** |
| Tilia | 57.89 | 40.74 | 47.83 | 23.90 | 188 |

## 5 Conclusions

In this paper, we introduced a novel dataset for multi-label and single-label tree species classification from aerial, S1 and S2 imagery. It includes more than 50,000 temporally (i.e., *same season and same year*) and spatially harmonized image patch triplets from the years 2011 to 2020 with a size of 60x60 m from all three sensors. The labels for 20 tree species (i.e., *15 tree genera*) were derived from forest administration data from Northern Germany.





Several DL and tree-based machine learning models using the dataset have been evaluated. With regards to the benchmarking results, we find that a pre-trained DL network using only the RGB bands of the 0.2 m resolution aerial data performs the best for modelling. The incorporation of additional spectral information from the S1 and S2 sensors led to a small increase in performance. We found that the utilization of airborne data performs sufficiently well for the study task, suggesting that high resolution RGB imagery provides enough information for DL models to differentiate forest stands at the tree species-level.

DL models only using aerial imagery reached high precisions (>80 %) for six of the 15 classes. This result demonstrates the potential to map predominant tree species at the 0.2 m resolution [c.f. Ahlswede et al., 2022]. However, our class-wise model performances using mono-temporal (i.e., *mono-seasonal summer scenes from different years*) data could not reach the accuracies of previous studies exploiting S2 multi-temporal (i.e., *multi-seasonal*) data [e.g., Immitzer et al., 2019; Grabska et al., 2019]. Nonetheless, we expect that our model approaches using multi-sensor fusion are robust to new data from different spatial areas.

Similarly to many forest remote sensing studies, our dataset contains a strong class imbalance which negatively affected the performance of under-represented classes. Future works based on the existing dataset could thus examine new methods for dealing with data imbalance. The simple weighting of the loss function based on class frequencies, which we applied, was not sufficient to overcome this issue. In addition, further studies can focus on improving the exploitation of the S2 multi-spectral data, as the models tested in this work were only capable of minor improvements over the RGB+NIR aerial imagery data.

For further improvements of the *TreeSatAI Benchmark Archive*, additional remote sensing products or label samples could be integrated. First, the integration of multi-seasonal data might disentangle further species-related information regarding phenology phases (e.g., spring flush, second flush, or autumnal leaf fall) or different branch structure types during the dormant phase (Kowalski et al. [2020]; Kollert et al. [2021]). This information could be derived from S2 and S1 image time series and additional aerial imagery from winter season. Second, active data with a higher spatial resolution (e.g., LiDAR) may improve the characterization of species-related tree crown features. Third, higher spectral resolution (i.e., *hyperspectral*) data, e.g. from the Environmental Mapping and Analysis Programme (EnMAP), may also reveal biophysical components differentiating tree species. Even though most of the aforementioned remote sensing products are not available for the same time frame, a combination could still be beneficial for tree species classification [Fricker et al., 2019; Weinstein et al., 2020; Mäyrä et al., 2021].

We highly encourage potential users of the *TreeSatAI Benchmark Archive* to create additional layers, generate further samples from other spatial domains or to develop new model architectures. This benchmark archive and its future applications can support in the provision of large-scale tree species maps, helping forest authorities to face challenges driven by global environmental changes.

*Code and data availability.* The *TreeSatAI Benchmark Archive* was made available through Zenodo (https://doi.org/10.5281/zenodo.6598390) [Schulz et al., 2022] under the *Creative Commons Attribution 4.0 International*. Full code examples are published on the GitHub repositories of the Remote Sensing Image Analysis (RSiM) Group (https://git.tu-berlin.de/rsim/treesat_benchmark) and the Deutsches Forschungszen-



trum für Künstliche Intelligenz (DFKI) (https://github.com/DFKI/treesatai_benchmark). Code examples for the sampling strategy can be
       made available by the corresponding author via email request.

*Author contributions.* All authors were involved in the conceptualization and methodology of the paper. Data investigation, compilation and dataset preparation was done by CS, SA, PH, BB, FA, JH, and BK. The analysis and code development was done by SA, CG, and JH. SA and CS wrote the original draft under the supervision of BD, BK and MF. All authors were involved in the review and editing process. The

project *TreeSatAI* was administrated by BK.

*Competing interests.* The authors declare that they have no conflict of interest.

*Financial support.* TreeSatAI was funded by the German Federal Ministry of Education and Research under the grant number 01IS20014A.

*Acknowledgements.* We are grateful to the Niedersächsische Landesforsten for providing us the aerial imagery and the reference data. We thank the European Space Agency for providing us the pre-processed Sentinel-1 and Sentinel-2 data and the respective software tools. Stenka

Vulova kindly improved our English.



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
