# Peer review of "TreeSatAI Benchmark Archive*: A multi-sensor, multi-label dataset for tree species classification in remote sensing"

_Earth System Science Data, 2022_

## Referee Comment (RC1)

Ahlswede et al introduce the TreeSatAI Benchmark Archive, a new data for tree species classification in Central Europe based on multi-sensor data from aerial, Sentinel-1 and Sentinel-2 imagery. This dataset contains labels of 20 European tree species (i.e., 15 tree genera). They also tested deep learning and machine learning models (residual neural networks (ResNet), multi-layer perceptron (MLP) or Light Gradient Boosting Machine (LightGBM) models) performances on this new dataset. This dataset is helpful to pre-train DL models for classifying species.

Overall, the manuscript conducted good work on data collection, statistic analysis, and results presentation. I think it is publishable if several minor issues can be addressed.

1. Figure 4 is not reader-friendly; the font is too small.

2. Introducing the dataset to the community is the main goal of this paper so that the author may include more information about the dataset. For example, in line 87, the description of the dataset itself is too concise.

---

## Author Response (AR1)

Author's response
Manuscript number: essd-2022-312

| # | Comments from Referees | Author's response | Manuscript changes |
|---|---|---|---|
| Anonymous Referee #1, 03 Oct 2022 | | | |
| 1 | Ahlswede et al introduce the TreeSatAI Benchmark Archive, a new data for tree species classification in Central Europe based on multi-sensor data from aerial, Sentinel-1 and Sentinel-2 imagery. This dataset contains labels of 20 European tree species (i.e., 15 tree genera). They also tested deep learning and machine learning models (residual neural networks (ResNet), multi-layer perceptron (MLP) or Light Gradient Boosting Machine (LightGBM) models) performances on this new dataset. This dataset is helpful to pre-train DL models for classifying species.

Overall, the manuscript conducted good work on data collection, statistic analysis, and results presentation. I think it is publishable if several minor issues can be addressed. | Dear referee,

Thank you for your helpful comments. In an updated version of the manuscript, we revised figure 4 to a more reader-friendly version and added more detailed descriptions of the dataset and its reference data. | Changes in the manuscript regarding your comments are highlighted in red. |
| 2 | Figure 4 is not reader-friendly; the font is too small. | We agree and changed the figure to a version with a larger font size. The readers can now better follow the forest inventory classes of the reference data that have been used for labeling. All other classes were excluded from the figure. | Figure 4 was replaced with a new figure version. |

| 3 | Introducing the dataset to the community is the main goal of this paper so that the author may include more information about the dataset. For example, in line 87, the description of the dataset itself is too concise. | We agree. The description of the dataset in the introduction was missing some important details. Therefore, we added a short information block about the dataset at the mentioned text passage and refer to a detailed description at the Zenodo publication. | Based on reference data from forest administration data in Germany, the dataset aims to gather multi-sensor and multi-label information for the classification of 20 tree species in Central Europe. The *TreeSatAI Benchmark Archive* consists of 50,381 image patches from aerial, Sentinel-2 (S2), and Sentinel-1 (S1) imagery (Fig. 1), with a range of 212 to 6,591 individual samples per class. All spectral bands and polarizations from the three sensor sources have been included. The patch sizes were harmonized to the same extent of 60x60 m. The S1 and S2 scenes were chosen as closely as possible to the season and years of the aerial imagery which was taken around August between the years 2012 to 2020. For a better reproducibility of our experiments, we created a fixed split of train (90%) and test (10%) data. Detailed information about the datasets' structure and version history can be found in the description of Schulz et al. [2022]. The data pre-processing and label derivation are further described in section 2.3 and 2.4. |

| Anonymous Referee #2, 27 Nov 2022 | | | |
|---|---|---|---|
| 4 | The manuscript presents a dataset for species classification and also answers various questions connected to the used technique/data combinations, which is important and interesting. However, I have some doubts about using Sentinel-1 for this purpose, as it was previously shown in many studies that S-1 only marginally improves accuracy. Secondly, when using only Sentinel-2 from the summer season, it is not surprising that classification accuracy improvement is limited. The majority of existing studies highlight the role of multi-temporal information (spring, autumn) from S-2 in species discrimination. You actually mention that finally in the conclusions. Maybe you can think about improving that process in the future, for example with seasonal metrics from S-2.

Still, your work is very valuable.

I have some minor comments: | We thank the anonymous referee for their comments.

Based on our findings, we agree that mono-temporal Sentinel-1 data provide limited information for the purpose of tree species detection. The same applies to some bands in the Sentinel-2 data.

We agree that future studies focusing on tree species classification would benefit from incorporating multi-temporal information.

The improvement of tree species classification through S2 time series or by including aerial imagery from spring, autumn or winter have been thoroughly discussed during the manuscript preparation. For future extensions of the benchmark archive, we will consider integrating S2 image time series (i.e., data cubes) containing the spectral and temporal dimensions for each sample. Through personal communication with the Public Agency for Geoinformation at Lower Saxony (LGLN), we found out that aerial photography is planned to be freely accessible in 2023. The data contains scenes from different seasons. We will also consider this data for extending the TreeSatAI benchmark archive in the future. | Changes in the manuscript regarding your comments are highlighted in blue. |
| 5 | Could you add what is the study area size/examined forest size? | We added the size of the study area to the study area section and we added the overall sum of forested area available for this study into the reference data section. | Ch. 2.1
[...] The study area, covering approximately 47,710 km², comprises both flat lands with a maritime climate (i.e., wet and |

| | | | winter-mild) in the North-West as well as low mountain ranges with a more continental climate (i.e., dry and winter-cold) in the South-East [Beck et al., 2018] [...]

Ch. 2.2
[...] First of all, from the federal state forest management data [...], a subset of 175,142 vector geometries, covering approximately 318,000 ha of forested area, was made available. [...] |
|---|---|---|---|
| 6 | Figure 4, particularly the a) part is difficult to read. And what is shown in the chart, the number of samples? | We agree. Another referee also mentioned the limited readability of Figure 4. The chart in the initial version of Figure 4 showed the number of available samples per class in the original reference data. The readers can now exactly follow the forest inventory classes of the reference data that have been used for labeling. All other classes were excluded. | Figure 4 was replaced with a new figure version. |
| 7 | Some methods used to assess accuracy are not described (or I couldn't find them). Maybe you could just briefly say something about what is weighted/micro, F1, and mAP score? | Thank you for pointing out this oversight on our part. We added a brief description of the metrics used to the introduction of chapter 4. The added description of weighted/micro averages should also help to clarify the reasoning behind why we chose these specific averages. | For all experiments, we trained models for a total of 150 epochs using a batch size of 32. For the learning rate, we use a cyclic learning rate scheduler [Smith, 2017] which modulates the learning rate value between 0.00005 and 0.001. In order to quantify model performance, we used recall, precision, F1 [Goutte and Gaussier, 2005], and mean average precision (mAP) [Everingham et al., 2010]. To |

| | | | achieve a single summary statistic across the multiple classes being predicted, an averaging technique must be applied to the aforementioned metric scores. We chose to apply both micro and weighted averaging. Micro takes the global average, which shows the performance of the model without taking class imbalance into consideration. In order to also evaluate model performance with respect to class frequencies, the weighted average was included. Here, a given metric is calculated for each class, and the weighted average of the class scores (with the weights being the total number of instances for each label) is calculated [Takahashi et al., 2022]. |
|---|---|---|---|
| 8 | Maybe you could put the subchapters in chapter 4 in the same order as the questions (or change the order of questions ;)) | We agree. The logical flow of the chapters between the questions and analysis steps was inconsistent. | The subchapters 4.2 and 4.3 have been swapped. |
| 9 | In lines 308-311 You refer to a study from Immitzer based on World-View data and I'm not sure if it makes sense as the difference between S-2 and World-view spatial resolution is huge… | Thanks for the comment, and indeed we do agree. Therefore, we have decided to remove this text passage from the manuscript. | We removed the mentioned text passage from the manuscript. |